# Genome-Wide Methylation Profiling in Canine Mammary Tumor Reveals miRNA Candidates Associated with Human Breast Cancer

**DOI:** 10.3390/cancers11101466

**Published:** 2019-09-29

**Authors:** Su-Jin Jeong, Kang-Hoon Lee, A-Reum Nam, Je-Yoel Cho

**Affiliations:** Department of Biochemistry, BK21 Plus and Research Institute for Veterinary Science, School of Veterinary Medicine, Seoul National University, Seoul 08826, Korea; godlovejsj13@snu.ac.kr (S.-J.J.); khlee02@snu.ac.kr (K.-H.L.);

**Keywords:** DNA methylation, canine mammary tumor, miRNA in cancer

## Abstract

Genome-wide methylation profiling is used in breast cancer (BC) studies, because DNA methylation is a crucial epigenetic regulator of gene expression, involved in many diseases including BC. We investigated genome-wide methylation profiles in both canine mammary tumor (CMT) tissues and peripheral blood mononuclear cells (PBMCs) using reduced representation bisulfite sequencing (RRBS) and found unique CMT-enriched methylation signatures. A total of 2.2–4.2 million cytosine–phosphate–guanine (CpG) sites were analyzed in both CMT tissues and PBMCs, which included 40,000 and 28,000 differentially methylated regions (DMRs) associated with 341 and 247 promoters of differentially methylated genes (DMGs) in CMT tissues and PBMCs, respectively. Genes related to apoptosis and ion transmembrane transport were hypermethylated, but cell proliferation and oncogene were hypomethylated in tumor tissues. Gene ontology analysis using DMGs in PBMCs revealed significant methylation changes in the subset of immune cells and host defense system-related genes, especially chemokine signaling pathway-related genes. Moreover, a number of CMT tissue-enriched DMRs were identified from the promoter regions of various microRNAs (miRNAs), including cfa-mir-96 and cfa-mir-149, which were reported as cancer-associated miRNAs in humans. We also identified novel miRNAs associated with CMT which can be candidates for new miRNAs associated with human BC. This study may provide new insight for a better understanding of aberrant methylation associated with both human BC and CMT, as well as possible targets for methylation-based BC diagnostic markers.

## 1. Introduction

Breast cancer (BC) is one of the most frequently diagnosed cancers in women, comprising 25% of total cancer in women [1]. Although many studies, projects, and clinical trials demonstrated the association between BC and some mutations on cancer driver genes such as *BRCA1/2* and *PIK3CA*, as well as familial history, only 5–10% of BC can be explained by genetic aberrations while the rest remains unclear [2].

Cancer studies using laboratory animal models, specifically rodents, were developed in diverse ways, such as the use of transgenic and targeted knockout mice [3]. However, it is difficult to mimic human environmental conditions using a laboratory animal model system. Companion animals such as dogs and cats, thus, became great model animals because of their closely shared environment with their owners [4]. Furthermore, canine mammary tumor (CMT) is a well-known animal model for BC because of its similarities in genetic, biological, and clinical features to BC [5]. Benign and malignant CMTs occur at a similar frequency, but classification was subdivided into malignant tumor subtypes by the World Health Organization (WHO). The 2011 classification subdivided CMTs into seven benign and 23 malignant subtypes, according to their prognostic values. Complex carcinoma is the most frequently diagnosed type of CMT (~20%), and it was subdivided into five subtypes (carcinoma complex type, carcinoma mixed type, carcinoma and malignant myoepithelioma, adenosquamous carcinoma, and intraductal papillary carcinoma) by the 2011 classification [6]. It is well known that the complex type of CMT, which includes a myoepithelial cell population and comprises ~20% of total CMT, is a great model for human myoepithelial cell-type BC, since myoepithelial cell-type BC is so rare in humans (~0.1%) [7].

Over the last decade, tumor immunity became a focus point due to the ability of immune systems to detect and destroy various cancer cells. Although the immune system mainly has a protective role, it does not always suppress tumor growth and dissemination successfully [8]. On the contrary, it was reported that tumor cells can co-opt immune cells such as macrophages into the tumor stroma, resulting in the promotion of endothelial cell proliferation in the tumor stroma and angiogenesis, which can enhance tumor cell mobility and invasiveness [9,10]. Moreover, platelets, tumor-associated macrophages, and regulatory T cells are known to increase the survival of circulating tumor cells, which promotes the establishment of metastatic foci [11]. Thus, understanding complex immune responses between tumors and the immune system is crucial to developing new diagnostic biomarkers and anti-cancer immunotherapy. 

Diverse environmental factors including nutrition, air pollution, and toxicants are involved in carcinogenesis through varying epigenetic alterations of DNA such as DNA methylation, histone deacetylation, and non-coding RNA regulation [12]. In particular, DNA methylation is highlighted due to its nature of early response to cancer [13]. The methylation profiles of the promoter regions of tumor suppressor genes such as *PTEN* and *BRCA2* were examined in different types of BC, and many current studies suggest that a few specific cytosine–phosphate–guanine (CpG) sites might be sufficient to provide prognostic information on BC with high accuracy and specificity [14,15]. Moreover, the Encyclopedia of DNA element (ENCODE) project revealed that the regulatory regions of genes such as promoters and enhancers are more conserved between human and dog models than human and rodent models. Indeed, a deeper understanding of methylation aberrations on gene regulatory regions is essential. 

MicroRNA (miRNA), of which the length is short (20–24 nucleotides long), is a well-studied non-coding RNA. Functionally, miRNAs mediate gene silencing via recognition of the 3′ untranslated region (UTR) of mRNAs, followed by guiding Argonaute proteins to these sites [16]. MicroRNAs can repress multiple mRNAs which harbor complementary sites through two post-transcriptional mechanisms, mRNA cleavage and translational repression. Almost 2600 mature human miRNAs are currently identified and many of them are conserved in vertebrates and invertebrates [17]. Moreover, numerous studies illustrated the function of oncogenic or tumor suppressive miRNA [18]. For example, highly expressed miR-34a promotes apoptosis [19], whereas miR-27a inhibits the expression of *ZERB10/RINZF*, which induces Sp factor expression followed by an increase in antipoetic and angiogenetic molecules [20]. Methylation is known as an important epigenetic regulator of miRNA expression in cancer [21], but miRNA studies across species are still veiled.

The current pilot study profiled methylation landscapes using reduced representative bisulfite sequencing (RRBS) in the mammary tumor and adjacent normal tissues, as well as peripheral blood mononuclear cells (PBMCs), of dogs. These comparative methylome data provide a biological link to CMT-specific differential methylation regions (DMRs) in both tissues and PBMCs. We suggest that this approach using the datasets of CMTs and matching normal tissues with PBMCs will provide new candidates for CMT biomarkers and help form a better understanding of human BC.

## 2. Results

### 2.1. RRBS Demonstrated CMT-Related Methylation Profiles in Tissues and Blood Cells

The sets of specimens, CMTs, adjacent normal tissues, and PBMCs, were collected from two individual patient dogs, and normal blood samples were drawn from dogs of matching breed in healthy condition. RRBS was performed on two sets of specimens obtained from dogs diagnosed as a carcinoma, complex type of CMT, for which degree of malignancy is low (grade I), to discover the cancer-specific methylation signature, and the result was validated by bisulfite conversion PCR in extended samples (Figure 1 and Table 1). Since a very limited number of samples were analyzed in the present study, we only focused on the carcinoma, complex type subtype and the Schnauzer dog breed in order to minimize variations. 

An averaged total of 2.9 gigabytes of data were sequenced, which covered 6.6–12.3% of canine genomic regions. Sequence statistics are summarized in Appendix A and the reads’ quality check is visualized in Appendix A. Assayed CpGs were mostly annotated in the intergenic and intron regions of the genome (97.9%), while ~2.2% were annotated in promoter and exon regions. The percentage methylation histogram was bimodal at both ends and the CpG base histogram showed typical shape of good quality in RRBS without PCR duplication (Appendix A). 

A CMT methylation signature was defined from a CMT-related DMR as a genomic region with a ≥15% change in methylation relative to matched normal with false discovery rate (FDR)-adjusted *q*-value ≤0.05. CMT methylation signatures were profiled in the comparison between CMTs and adjacent normal tissues and between PBMCs obtained from CMT patient and healthy dogs. A total of 41,044 and 27,667 DMRs were identified from the comparison of tissues and PBMCs, respectively. As known, genomic DNA in cancer cells when compared to normal cells is hypomethylated in human; in concordance, hypomethylation (26,562 in tissues and 17,631 in PBMCs) was found almost twice as often hypermethylation (14,482 in tissues and 10,036 in PBMCs) in CMT-related DMRs. Since the two healthy dogs consisted of both male and female sex, we computed CMT-related DMRs excluding DMRs on the X chromosome (Appendix A).

Hierarchical clustering and heatmap analysis using the CMT-related DMRs successfully differentiated PBMCs in healthy condition from those in CMT conditions, as well as CMT tissues from adjacent normal tissues (Figure 2A,B). 

The list of CMT-related DMRs for which methylation level was significantly changed in CMTs only is summarized in Appendix A. It will be useful to investigate epigenetic alterations noticed from cancer liquid biopsies using cell-free DNA (cfDNA). 

To better understand the methylation landscape of CMT, all DMRs were separated according to genomic distribution in both gene and intergenic regions. We identified that DMRs were 35–38% hypermethylated and 62–65% hypomethylated in tissues, while 27–38% were hypermethylated and 62–73% hypomethylated in PBMCs. The genomic distribution of DMRs is depicted by volcano plot in Figure 2C,D, and the total number of DMRs is summarized in Appendix A. The largest number of DMRs was observed in intergenic regions, followed by intron and promoter regions. These data were consistent with a previous study [22]. The number of DMRs found in tumor tissues was higher than in PBMCs. However, the percentage distribution of hyper- and hypomethylated DMRs was similar between tissues and PBMCs (35% and 36% were hypermethylated in CMT tissues and PBMCs, respectively). Unexpectedly, more than half of the assayed region was obtained from the intergenic region after restriction enzyme digestion enrichment for CpG island (CGI)regions. This result could be explained by reports of canine intergenic CGI density being higher than that in the human genome [23]. We also compared CG content in the intergenic DMR regions determined by chromosomal average. CG content of assayed intergenic DMRs was ~60%, which is higher than the chromosomal average (~40%), and means that CGI enrichment was successful (Appendix A). On the other hand, more than half of the promoter DMRs were primarily hypomethylated in CMTs when compared to matched normal. Additionally, intron regions were also significantly enriched in DMRs. This may suggest that methylation dynamically regulated gene expressions at not only the promoter region but also the intron region. 

### 2.2. Differentially Regulated DNA Methylations in CMT Tissues and PBMCs

DMGs are defined as genes existing within 2 kb up- and downstream of a DMR. A total of 341 DMGs in tissues (202 hypo- and 139 hypermethylated) and PBMCs (175 hypo- and 72 hypermethylated) were subjected to gene ontology (GO) and Kyoto encyclopedia of genes and genomes (KEGG) pathway analysis (Figure 3, Appendix A).

Cancer-associated terms in GO enrichment analysis are illustrated in Figure 3A,B. In tissues, DMGs were significantly enriched in biological processes (BP) of extrinsic apoptotic signaling pathway via death domain receptors (GO:0008625), positive regulation of protein kinase B signaling (GO:0051897), and positive regulation of cell proliferation (GO:0008284) (Figure 3A). This is noticeable because a different study reported that protein kinase B signaling, including *IGF2*, *TNF*, and *ZP3*, has activity that inhibits apoptosis in human BC via acting growth factor transducer [24]. The terms of hypomethylated DMGs enriched in positive regulation of cell proliferation and *KRT* and *MMP* families that involved tumor metastasis promoting through extra cellular matrix (ECM) remodeling [25] were also hypomethylated (Appendix A). Hypermethylated genes were enriched in positive regulation of cytosolic Ca^2+^ concentration (GO:0007204), negative regulation of K^+^ transmembrane transport (GO:1901380), and positive regulation of apoptotic process (GO:0043065) (Figure 3A). Aberrant methylation of K^+^ transmembrane transport-related genes, a frequently observed feature in human cancer, was also hypermethylated in CMTs [26]. *CAV1/3* and *KCNH2* were enriched in this term; the *CAV1* gene is thought to function as a tumor suppressor or modifier gene in mammary epithelia [27].

In PBMCs, hypomethylated genes were highly clustered in positive regulation of cytosolic Ca^2+^ concentration (GO:0007204), cell–cell signaling (GO:0007267), and macrophage chemotaxis (GO:0048246). Hypermethylated genes were enriched in negative regulation of apoptotic process (GO:0043066) and cell growth (GO:0016049) (Figure 3B). It is an interesting feature that positive regulation of cytosolic Ca^2+^ concentration GO was hypomethylated in cancer PBMCs while it was hypermethylated in cancer tissues.

Moreover, through KEGG pathway analyses, we determined that the immune response-related pathways were hypomethylated in both CMT tissues and PBMCs (Figure 3C,D). Interestingly, cytokine–cytokine receptor interaction was hypomethylated and was likely to be enriched in CMT tissues, indicating that cytokine–cytokine receptor interaction genes are demanded for crosstalk in immune cells. On the other hand, chemokine signaling was also enriched in PBMCs (Figure 3C). Aberrant regulation of chemokine signaling via DNA methylation dictated that many signaling pathways in immune cells are affected in the tumor environment. Although many studies were conducted on cancer immune systems, there is limited knowledge on the methylation status of specific gene sets directly associated with cancer cells.

### 2.3. Cancer-Associated miRNA Expressions Regulated by Methylation in CMT

One interesting finding in the KEGG pathway enrichment analysis was that the term of microRNAs in cancer (KEEG:05206) was highly enriched in both hypo- and hypermethylated DMGs in CMT tissues, and hypermethylated DMGs in PBMCs (Figure 3C,D). Importantly, we found that the lists of miRNAs enriched in each hypo- or hypermethylated DMG were clearly separated according to their roles in cancer, such as oncogenic and tumor suppressive miRNAs. Many of the identified miRNAs, including miR-141, miR-142, and miR-10B, which are all known as cancer-associated miRNAs, were enriched in hypomethylated DMGs. On the other hand, miR-22, miR-149, and miR-124-2, which are all reported with cancer suppressive activity, were found in hypermethylated DMGs [28,29,30,31]. These results suggest that cancer-enriched methylation profiles may directly link to cancer development via regulating cancer-associated miRNA expression (Table 2). 

By extending searches on miRNAs involved in tumorigenesis, we identified a total of 9179 (2996 hyper- and 6183 hypomethylated) DMRs associated with miRNA from RRBS (Appendix A). Out of 9179 regions, 121 DMRs were located in the miRNA promoter region. Hierarchical clustering showed that most miRNA-associated DMRs were hypomethylated in tumors (Figure 4A). 

Of note, many of these miRNAs associated with DMRs were shared by miRNAs found in human BC studies (Figure 4B, Table 2). 

We selected two miRNAs (cfa-miR-96 and 149) through a literature survey to identify and confirm BC-related miRNA expression. According to previous studies in human, has-miR-96 is known to be a oncomir, and has-miR-149 [45,46] is known to be a tumor-suppressive miRNA. Validation by quantitative real-time PCR (qRT-PCR) was performed on an independent set of samples with complex carcinoma patient tissues. Primer sequences are shown in Appendix A. The expression of cfa-miR-96, an oncogenic miRNA in human, was significantly increased in CMTs, which is in agreement with its promoter hypomethylation, whereas expression of cfa-miR-149, tumor-suppressive miRNA, was decreased in CMTs, consistent with its promoter hypermethylation (Figure 4C, Appendix A). 

To further understand the molecular mechanism of these miRNAs, we searched for the targets of these miRNAs by using the sequence-based target prediction programs TargetScan (http://www.targetscan.org) and miRDB (http://mirdb.org/miRDB) [47,48]. Putative target genes were selected by using TargetScan and our gene expression profiles in CMT. Among the top five predicted target genes of cfa-miR-96, three genes, *BRPF3, ADCY6*, and *LRIG1*, were significantly downregulated in tumors compared to normal tissues (Figure 4D). On the contrary, *RNF2*, a predicted target gene of tumor-suppressive miRNA cfa-miR-149, was upregulated in tumor tissues (Figure 4D).

### 2.4. Potential Candidates for New BC-Associated miRNA cfa-miR-8832

Among all new miRNA candidates (Appendix A), cfa-miR-8832, one of the miRNAs newly identified in the present study, was significantly upregulated in CMT (Figure 5A). 

To determine the potential role of cfa-miR-8832 in cancer, we surveyed target genes and their gene expression levels in tumor and normal tissues. Five potential target genes including *STIM1* and *GNAO1* were found, and their gene expression was significantly decreased in tumor samples (Figure 5B,C and Appendix A). *GNAO1* is known as a tumor suppressor gene in some human cancers [49], and its expression in CMT was downregulated. We then demonstrated that cfa-miR-8832 is a candidate of human BC-associated miRNA through the conservancy existing between these two orthologous target genes. Sequence homology of UTR regions of these two target genes, which cfa-miR-8832 acts on, was compared in human and dog and had highly conserved miRNA targeting sequences in both species (Figure 5C, Appendix A). This result suggests that the comparative approach using epigenetic regulation such as methylation on miRNA regulatory regions in CMT can be a valid approach for the investigation of novel biomarker and target miRNAs and genes.

### 2.5. Aberration of CpG Methylation on the Promoter Regions of *CYP21A2*, *KCNH2*, *KRT5*, and *MMP7* Genes as Methylation Biomarkers for CMT

The DMRs identified from gene promoter regions by RRBS was validated by using bisulfite direct- or cloning-based sequencing (BSC) in additional clinical specimens (Figure 6).

Detailed methods including primer sequences and product sizes are shown in Section 4 and Appendix A. The promoter methylation of *KCNH2* and *CYP21A2* genes, which are known as a regulator of apoptosis and associated adrenal hyperplasia, was measured by BSC in the three pairs of CMTs and adjacent normal samples. *KCNH2* and *CYP21A2* promoter regions consisting of 10 (chr16: 15,042,868–15,043,044) to 12 CpG sites (chr12: 1,451,023–1,451,292) were drastically hypermethylated in cancer samples when compared with normal (Figure 6A,B). In addition, promoter hypomethylation of *KRT5* and *MMP7* genes was confirmed in cancer samples. Methylation quantification by BSC in the gene promoters, 189 bp of *KRT5* (chr27: 2,567,252–2,567,440) and 306 bp of *MMP7* (chr5: 29,187,687–29,187,992), confirmed hypomethylation of CpGs in CMT (Figure 6C,D).

## 3. Discussion

Recently, numerous genome-wide cancer methylome studies were performed in various types of human cancers [50]. Canine mammary tumor (CMT) is a good animal model for human breast cancer and was studied in genetic, biological, and clinical settings of comparative medicine [51]. Even though CMT and human BC share many pathological features, previous comparative oncology studies focused on histological and molecular subtypes according to hormone receptors.

In this study, genome-wide methylation was profiled in the CMT and adjacent normal tissues coupled with matching PBMCs obtained from two Schnauzer dogs diagnosed with carcinoma, complex type CMT. In the present study, we analyzed methylation profiles in the least number of samples and only within one subtype of CMT (carcinoma, complex type) and dog breed (Schnauzer); thus, it might not be easy to generalize our findings to overall CMT or human BC. Further analysis with increased sample size and different types of CMT is necessary. However, this study focusing on complex type carcinoma in one dog breed successfully revealed CMT-associated methylation signatures for at least the complex type. As expected, the majority of DMRs identified in CMT were hypomethylated. This genome-wide loss of DNA methylation is a frequently observed feature in various human cancers [52]. 

We also profiled DNA methylation in PBMCs. It is well known that PBMCs are a heterogeneous collection of different cell types, composed of very differently methylated DNA and changed in various human diseases including cancer. Indeed, accounting for the cellular heterogeneity of PBMCs might be important for better interpretation of epigenome-wide association results. We, thus, tried to lift over CMT-enriched differentially methylated cytosines (DMCs) to human blood cell type-specific differential methylation data because cell type-specific methylation is yet to be demonstrated for dog [53]. Using the top 50 DMCs for each cell type, the alteration of cell population during tumorigenesis was estimated (Appendix A). The cell populations of B and CD4 T cells were increased by 56.2% and 69.9%, respectively, in PBMCs of CMTs. On the other hand, monocyte and natural killer (NK)cell populations were slightly decreased in PBMCs of CMTs. This result suggests that DMCs identified in PBMC of CMT might be affected by changed numbers of certain cell types. The cell type-specific methylation profiling in blood will be guaranteed in sorted cell populations and in both healthy and CMT. On the other hand, DNA methylation profiles in PBMCs or whole blood are more applicable from the diagnosis point of view.

The current study successfully revealed CMT-enriched DMRs in both tissues and PBMCs, and the putative roles of DMRs were characterized by the GO and pathway analysis of associated genes. As expected, many apoptosis-related genes, involving *ARHGEF2*, *TNFRSF12A*, and *SFRP2*, were hypermethylated in CMT, and some oncogenes in human cancers, *HRAS*, *FAM83H*, and *RET*, were found as hypomethylated [54,55,56]. According to KEGG analysis in PBMCs, hypomethylated genes were enriched in chemokine signaling pathway (04026) and calcium ion signaling pathway (04020). It is known that the calcium ion is a secondary messenger for lymphocytes, and increased intracellular calcium levels are necessary for cytoskeleton rearrangement, leading to T-cell-dependent responses [57]. Together, these results suggest that T cells are activated by chemokine and Ca^2+^ ion signals in the tumor environment.

MicroRNA dysregulation is a hallmark feature in tumorigenesis. However, aberrant methylation-induced miRNA dysregulation in CMT remains to be elucidated. In this study, for the first time, we demonstrated that the terms of miRNA in canine cancer were highly enriched in KEGG pathway, validated by the RNA expressions of target genes, as well as itself, in CMT. Our results confirmed that two miRNAs, cfa-miR-96 and -149, which are known as oncogenic and cancer-suppressive, respectively, and their target genes are highly conserved in dog and human. This result indicated that miR-96 and 149 are common onco- and cancer-suppressive miRNAs in several species. Furthermore, we newly identified cfa-miR-8832 in the present study as an miRNA associated with both CMT and human BC. Among cfa-miR-8832 target genes, *GNAO1* (guanine nucleotide-binding protein-alpha O1) is known as a tumor suppressor gene in some human cancers [49] and its expression in CMT was downregulated. *GNAO1*, containing a GTP hydrolysis domain, is a subunit of GPCRs (G-protein coupled receptors), involved in signal transductions such as the cAMP pathway, and K+ and Ca2+ channels. In human BC, an R243H mutation of *GNAO1* is involved in tumorigenesis through oncoprotein formation [58]. Another previous study reported that *GNAO1* is able to reduce cell proliferation and induce senescence in human hepatocellular carcinoma (HCC) [49]. According to Kaplan Meier plot of six aberrantly methylated genes in tissue, low expression of *GNAO1* correlated with breast cancer. (Appendix A) All these results indicate that uncontrolled regulation of *GNAO1* may have a relationship with tumorigenesis, as well as being a potential biomarker in both human and canine. Furthermore, the 3’ UTR of *GNAO1* consists of several loci for miRNAs, and it will be very important to further investigate which miRNAs and recognition sites are key regulators for *GNAO1* expression (Figure 5C, Appendix A). Six more miRNAs, has-miR-7113-5p, -6837-5p, -1915-3p, -4685-5p, -6764-5p, and -6798-5p, were also identified around the seed sequences of cfa-miR-8832 (CCCCUGG) in the *GNAO1* gene, and all of these miRNAs showed anti-correlation between survival rate and their expression. Kaplan meier (KM) plot analysis showed that, somehow, these miRNAs redundantly act on *GNAO1* together (Appendix A). On the other hand, another target gene, *STIM1*, showed a different expression pattern in human BC (upregulated) [59] than in CMT (downregulated). It needs to be investigated further, but this result indicates that there is a different regulation mechanism of cfa-miR-8832 acting on *STIM1* expression in human BC and CMT. 

In addition, we also validated several DMRs around gene promoters of *KCNH2*, *CYP21A2*, *KRT5*, and *MMP7* using BSC-targeted sequencing. These were chosen based on their gene expression patterns in cancer according to the literature survey: *KRT5* and *MMP7* for increased gene expression in cancer [60], *KCNH2* for ambiguous expression in cancer [61], and *CYP21A2* for unknown expression in BC. The *CYP21A2* gene, which encodes a steroid 21-hydroxylase, is known as a gene associated with congenital adrenal hyperplasia, and very little is known about it in BC. Targeted sequencing after BSC confirmed the RRBS result of differential methylation of the target gene promoter regions between normal and cancer samples. Hypomethylation was found in both *KRT5* and *MMT7* genes in cancer, in concordance with high gene expression in cancers. On the other hand, *CYP21A2* and *KCNH2*, for which the expressions were not clearly investigated in human cancer, showed a heavily methylated promoter in cancer compared with a rarely methylated one in normal cells. It suggests that aberrant methylation of these gene promoter regions can be candidates for diagnostic markers in both CMT and human BC. 

## 4. Materials and Methods

### 4.1. Specimens

This study was reviewed and approved by the Seoul National University Institutional Review Board/Institutional Animal Care and Use Committee (IACUC SNU-170602-1, approval 26 July 2016), and all methods were performed in accordance with the relevant guidelines and regulations. Dogs of the Schnauzer breed diagnosed with carcinoma, complex type CMT and healthy controls designated symptomless based on the results of blood chemistry were enrolled in this study. For RRBS in the discovery phase, two sets of specimens consisting of CMTs and adjacent normal tissues with blood were collected from CMT patient dogs, and control blood samples were isolated from healthy dogs of the same breed. Three sets of tissue were used for bisulfite conversion PCR and cloning-based sequencing in the validation phase. The overall process is depicted in Figure 1 and listed in Appendix A.

### 4.2. PBMC Isolation

PBMCs were isolated from whole blood by using a standard, previously validated protocol. Briefly, blood was applied to an equal volume of Ficoll-Paque™ PLUS (GE Healthcare, Orsay, France) and centrifuged for 30 min at 500× *g*, 18 °C, without a break. PBMCs were obtained from the central white band of the gradient. Samples were stored at −80 °C in freezing medium (90% FBS, 10% DMSO) until use.

### 4.3. Nucleic acid Isolation

Genomic DNA was extracted from four canine mammary tissues and four PBMCs using the DNeasy blood and tissue kit (QIAGEN, Hilden, Germany). Genomic DNA was eluted with 30 μL of elution buffer. Total RNA was isolated from six cryopreserved canine mammary tissues using TRIzol reagent (Thermo Fisher Scientific, Inc. Waltham, MA, USA) and treated with RNase-free DNase I (Enzynomics, Dae jeon, Korea) before reverse transcription at 37 °C for 30 min and DNase1 inactivation by phenol–chloroform extraction. All methods were performed according to the manufacturer’s instructions. The RNA was checked via 2% agarose gel electrophoresis to observe the 18S and 28S ribosomal RNA (rRNA) band integrity. The purity and quality of extracted samples were assessed by OD (260:280) ratio, using a microplate reader (Epoch 2; BioTek Instrument, Inc., Winooski, ST, USA), and stored in a deep freezer until use.

### 4.4. RRBS and Genome-Wide Methylaion Profiling

To construct the *Msp*I- and *ApeK*I-digested RRBS library, 500 ng of input genomic DNA was assembled into 50-μL reactions with *Msp*I (New England Biolabs, Ipswich, MA, USA). After digested DNA purification, methylated adaptor ligation and size selection (160–24bp) were sequentially performed. Bisulfite conversion was conducted using a ZYMO EZ DNA Methylation-Gold Kit™ (ZYMO research, Irvine, CA, USA), following the manufacturer’s instructions. The final libraries were generated by PCR amplification using PfuTurbo Cx Hotstart DNA polymerase (Agilent technologies, Santa Clara, CA, USA). RRBS libraries were analyzed with an Agilent 2100 Bioanalyzer (Agilent Technologies, Santa Clara, CA, USA). Sequence data were preprocessed by the following steps: potentially existing sequencing adapters and raw quality bases in the raw reads were trimmed using Skewer [62]. The cleaned high-quality reads were mapped to the reference genome with three-letter converted genome by the bs_seeker2-align module of BSseeker2 software [63]. Differentially methylated regions (DMRs) were identified using methylKit [64], which uses the BSseekers2 output data as an input. We briefly assayed a tiled genomic region of 200-bp window size and calculated methylation level (methyl cytosine/(methyl cytosine + unmethyl cytosine)) in the tiled region. Methylation difference was calculated between tumor and adjacent normal control. Chi-square test was used as a default setting to calculate *p*-value; then, *p*-value was adjusted to *q*-value using the sliding linear model [65]. After *q*-value calculation, DMRs were filtered by CpG coverage of ≥10, percentage methylation difference ≥15, and *q*-value ≤0.05. RRBS and DMR findings were performed in LAS Bioscience (LAS Corporation, Gimpo, Korea).

### 4.5. Bisulfite Conversion PCR and Sequencing

A 500-ng sample of high-quality genomic DNA was subjected to bisulfite conversion using an EZ DNA Methylation-Lightning Kit according to the manufacturer’s protocol (Zymo Research, Irvine CA, USA). Primer sets were designed for direct bisulfite sequencing validation to confirm methylation status using MethPrimer [66], and they are listed in Appendix A. During PCR, uracil is converted to thymine and the product is amplified. Amplicons were eluted from the agarose gels using QIAquick Gel Extraction Kit (Qiagen, Hilden, Germany). Sequencing was performed in Solgent Co. Ltd. (Solgent Co. Ltd., Seoul, Korea). 

### 4.6. Stem-Loop RT-qPCR

The stem loop RT and qPCR primers for miRNAs were designed using the web-based designing tool at http://genomics.dote.hu:8080/mirnadesigntool/ and synthesized (Bioneer, Daejeon, Korea) (Appendix A). The six miRNAs selected for this study and endogenous control miRNA were reverse-transcribed using each miRNA-specific stem-loop RT primer. Complementary DNA (cDNA) was generated from 500 ng of total RNA using an Omniscript™ RT kit (Qiagen, Hilden, Germany) according to standard procedures. qPCR was performed using the following SYBR protocol. qPCR was performed with a CFX96 Real-Time PCR system (Roche Applied Science, Rotkreuz, Switzerland) and quantitation cycle (Cq) values were normalized to *RPS* gene, which was used as an internal control. The amplification profile was as follows: one cycle of denaturation at 95 °C for 10 min, followed by 50 cycles of denaturation at 95 °C for 30 s, annealing 58–60 °C for 30 s, and extension at 72 °C for 30 s. All experiments were performed in triplicate.

### 4.7. Gene Ontology (GO) and Pathway Analysis

GO and KEGG pathway enrichment analyses were performed using DAVID bioinformatics resources [67] and Cytoscape 3.70 [68] to predict the gene function of identified differentially methylated genes (DMGs). A DMG was defined by an existing gene within 2 kb up- and downstream of the DMR. 

### 4.8. Target Prediction of miRNAs

In order to identify the potential association between mRNAs and miRNAs, we used web-based target prediction tools miRDB and TargetScan.

### 4.9. Statistical Analysis

Expression levels of individual miRNA of paired samples were determined by the ddCt approach. Expression and bisulfite sequencing data used tumors and normal tissues; the results are expressed as means ± standard error of the mean (SEM), and the statistical significance was assessed by two-tailed paired Student’s *t*-test using GraphPad (La Jolla, California, USA). Asterisks indicate * *p* < 0.05, ** *p* < 0.01, and *** *p* < 0.001. DMRs were filtered by *q*-value, coverage, and methylation difference, followed by clustering with average linkage and Pearson correlation. 

## 5. Conclusions

Taken together, this study revealed a CMT methylome in both tissues and PBMCs using RRBS, and showed CMT-enriched aberrant methylations; we newly demonstrated that differential methylation in CMT regulates cfa-miR-8832 expression, which may block tumor suppressive target gene expression. In addition, this study also provided the regions of hypo- and hypermethylated genes in CMT that may provide a highly valuable resource for the investigation of the epigenetic regulation of human breast cancers.

## Figures and Tables

**Figure 1 cancers-11-01466-f001:**
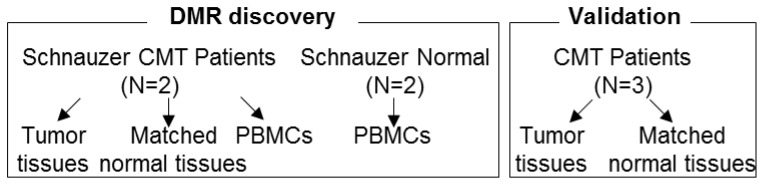
Scheme of research flow and specimens. DMR: differentially methylated region, PBMC: peripheral blood mononuclear cell, RRBS: reduced representation bisulfite sequencing.

**Figure 2 cancers-11-01466-f002:**
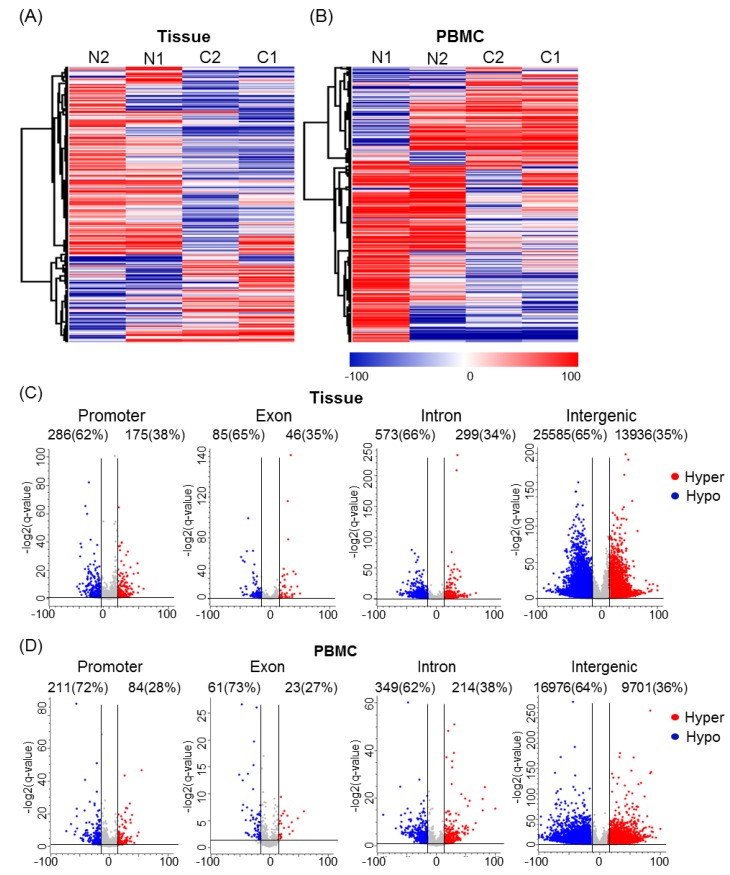
Genome-wide DNA methylation changes in CMT tissue and PBMCs. (**A**,**B**) Hierarchical clustering and heatmap of DMRs defined by difference ≥15%, false discovery rate (FDR)-adjusted *q*-value ≤ 0.05, and read coverage ≥10 in CMT tissues. The color indicates the percentage methylation: hyper (in red) and hypo (in blue). N: normal, C: CMT. Heatmaps for tissues (**A**) and PBMCs (**B**) were generated by complete linkage clustering using a Pearson correlation. (**C**,**D**) Volcano plots of all DMRs separated according to their genomic features. The *x*-axis represents methylation percentage and the *y*-axis represents −log2 (*q*-value). Methylation changes in promoter, exon, intron, and intergenic regions were examined; DMRs are depicted as blue (hypo) or red (hyper) dots in CMT tissues (**C**) and PBMCs (**D**). CMT: canine mammary tumor; PBMC: peripheral blood mononuclear cells; DMR: differentially methylated region.

**Figure 3 cancers-11-01466-f003:**
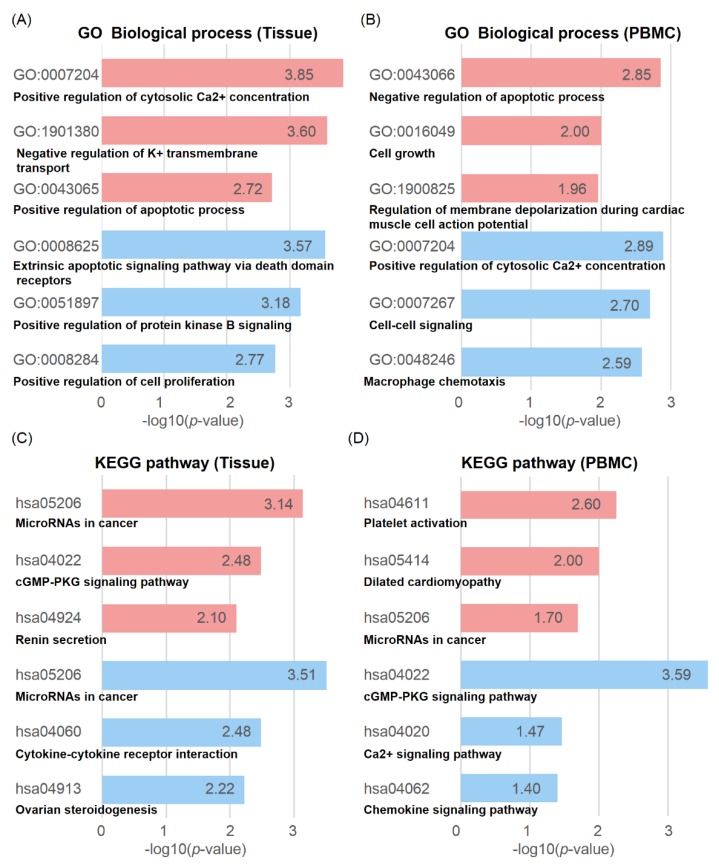
Enrichment analysis of gene ontology (GO) and KEGG pathway using DMGs in CMT. Gene ontology (GO) and KEGG pathway analysis in tissue and PBMC DMGs are depicted with −log (*p*-value), where the *y*-axis represents terms and IDs. From tissues, 139 hyper- and 202 hypomethylated DMGs, and, from PBMCs, 72 hyper- and 175 hypomethylated DMGs were applied. The top three significant GO_BP terms are depicted. Colors represent hyper (red) and hypo (blue) methylations. GO analysis in biological processes in tissues (**A**) and in PBMCs (**B**). KEGG pathway analysis in tissues (**C**) and PBMCs (**D**). KEGG: Kyoto encyclopedia of genes and genomes; DMG: differentially methylated gene; BP: biological process; CMT: canine mammary tumor; differentially methylated genes.

**Figure 4 cancers-11-01466-f004:**
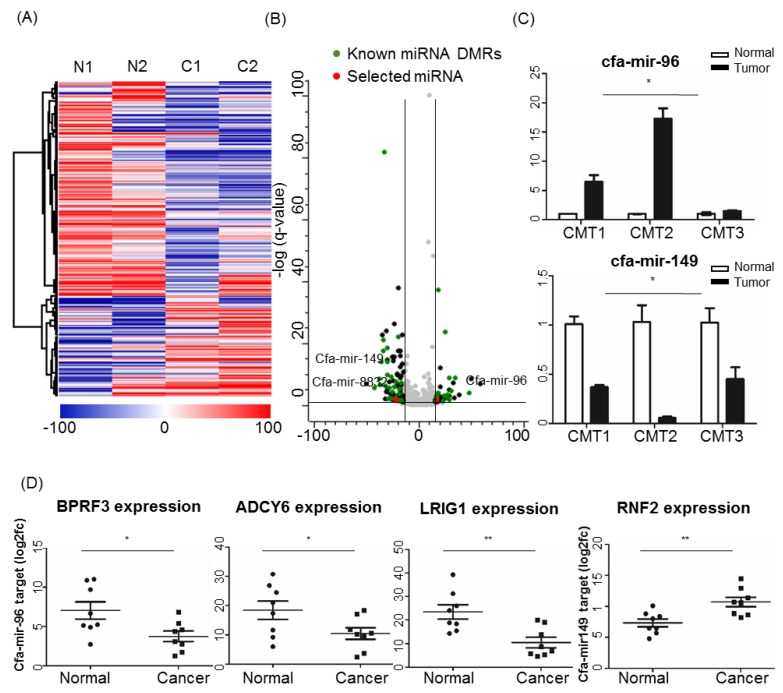
Differential miRNA methylation study in CMT samples. (**A**) Hierarchical clustering and heatmaps of DMRs on miRNA promoters were generated by complete linkage clustering using a Pearson correlation. Color scale represents percentage methylation. Hyper- (in red) and hypomethylation (in blue), N1, N2: normal, C1, C2: CMT. (**B**) Volcano plot presenting methylation differences in miRNAs consisting of known and new miRNA genes. Known miRNAs in green, new miRNAs in black; red indicates being chosen for validation, and gay is not significant. (**C**) Cfa-mir-96 and cfa-mir-149 expression levels determined by relative RT-qPCR in CMTs and adjacent normal tissues. The *y*-axis represents the log2 fold change value. (**D**) Dot plots of miRNA-targeted gene expression determined by RNA sequencing (RNA-seq) data in CMTs and adjacent normal tissues. Asterisks indicate * *p* < 0.05 and ** *p* < 0.01. miRNA: micro RNA; CMT: canine mammary tumor; DMR: differentially methylated region.

**Figure 5 cancers-11-01466-f005:**
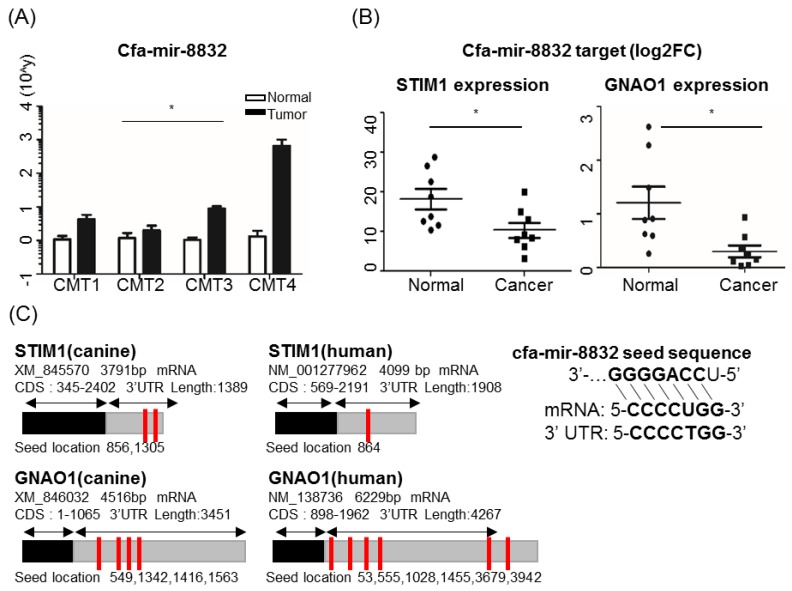
Canine mammary tumor (CMT)–associated miRNA, cfa-mir-8832 expression inversely correlated with the expression of putative target genes, *STIM1* and *GNAO1*. (**A**) Cfa-mir-8832 expression in CMT and adjacent normal tissues (* *p* < 0.05). The *y*-axis represents the 2^Ct^ value. (**B**) Dot plots of *STIM1* and *GNAO1* gene expression in CMT and adjacent normal tissues. *RPS* gene was used as the normalization gene. The *y*-axis represents log2 fold change value. (**C**) Putative miRNA target sites on *STIM1* and *GNAO1* orthologous genes in dog (left) and human (right). Gray indicates 3’ untranslated regions. Red lines indicate target loci of miRNA cfa-mir-8832. GGGGACC: seed sequence of cfa-mir-8832.

**Figure 6 cancers-11-01466-f006:**
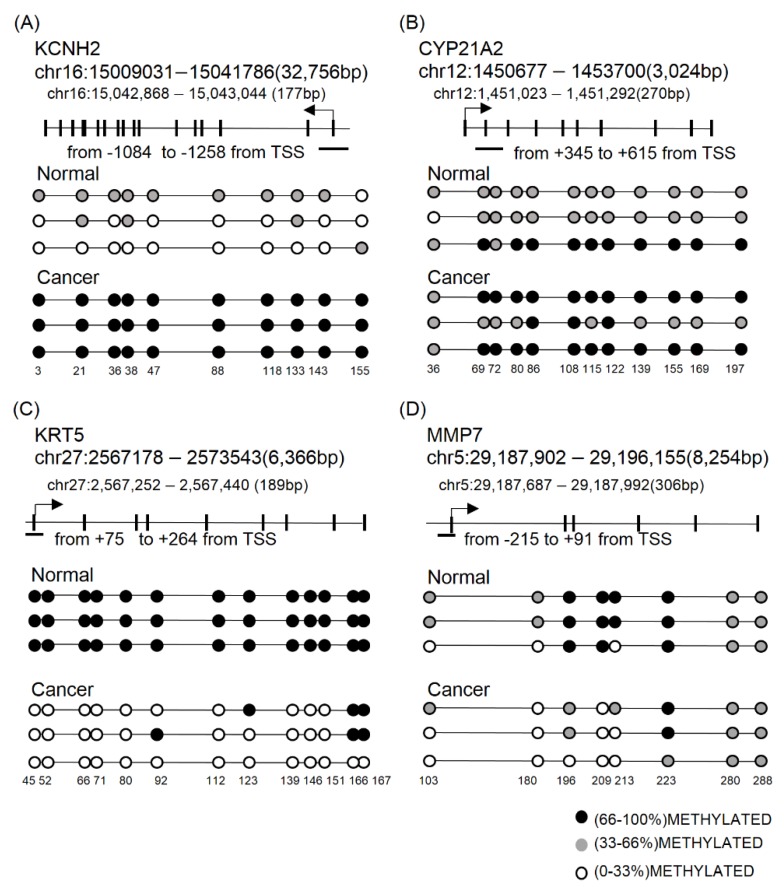
Validation of RRBS with targeted sequencing after BSC. (**A**,**B**) DMRs on the promoter region of four target genes was selected for validation. Various numbers of CpG sites on each promoter region are represented by a lollipop plot. *KCNH2* (**A**) and *CYP21A2* (**B**) genes were confirmed by cloning-based sequencing after BSC. Methylation states of each CpG site were measured by ratio of converted C to T after BSC PCR in five different clones. (**C,D**) Methylation on the promoter regions of *KRT5* (**C**) and *MMP7* (**D**) genes was measured by direct sequencing after BSC PCR. Signal density of C and T on each CpG site was calculated as methylation state and is depicted by color scale in three groups (66–100%: black, 33–66%: gray, and 0–33%: white). Bottom numbers indicate CpG loci based on transcription start site of gene. RRBS: reduced representative bisulfite sequencing; BSC: bisulfite conversion.

**Table 1 cancers-11-01466-t001:** Patient information.

Breed	Sex	Age	Diagnosis	PBMC	Tissue
Schnauzer	FS	13	Carcinoma complex type	O	O
Schnauzer	FS	9	Carcinoma complex type	O	O
Schnauzer	MN	10	Healthy	O	O
Schnauzer	FS	10	Healthy	O	O
Maltese	F	12	Carcinoma complex type	-	O
Dachshund	FS	14	Carcinoma complex type	-	O
Dachshund	F	3	Carcinoma complex type	-	O

Clinical characteristics of complex carcinoma and normal canine samples used in this study. Tissues are composed of canine mammary tumors (CMTs) and adjacent normal tissues. PBMC: peripheral blood mononuclear cell; FS: female spayed; MN: male neutralized; F: female.

**Table 2 cancers-11-01466-t002:** Potential functions of different microRNAs (miRNAs) in cancer progression.

Name	Methylation Status in Canine	Function in Human	References
miR-142	Hypo	Oncogenic	[32]
miR-141	Hypo	Oncogenic	[33]
miR-200C	Hypo	Oncogenic	[33]
miR-96	Hypo	Oncogenic	[34]
miR-10B	Hypo	Oncogenic	[35]
miR-184	Hypo	Oncogenic	[36]
miR-33A	Hypo	Oncogenic	[37]
miR-95	Hypo	Oncogenic	[38]
miR-203	Hypo	Oncogenic	[39]
miR-22	Hyper	TSG	[40]
miR-149	Hyper	TSG	[41]
miR-124-2	Hyper	TSG	[42]
miR-34C	Hyper	TSG	[43]
miR-182	Hyper	TSG	[44]

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
