# Peer review of "Genome-Wide Methylation Profiling in Canine Mammary Tumor Reveals miRNA Candidates Associated with Human Breast Cancer"

_cancers, 2019, doi:10.3390/cancers11101466_

Round 1

Reviewer 1 Report

In this study, authors have used only 2 replicates per group, which is a very very low sample size considering the cost of experiments, to generalize on the results found in the study. Authors are highly recommended to increase the sample sizes to make the results stronger to be trusted by the scientific community.

The method sections of manuscript have not been adequately described. For example in section 4.3, they have performed both DNA and RNA isolation. Why the title is only Genomic DNA Isolation?

For the hierarchical clustering and heatmaps (2A, 2B and 4A), what is type of data and method used for clustering is not clear. Again showing only two replicates per group can be perfect compared to showing multiple replicates.

Author Response

Q1: In this study, authors have used only 2 replicates per group, which is a very very low sample size considering the cost of experiments, to generalize on the results found in the study. Authors are highly recommended to increase the sample sizes to make the results stronger to be trusted by the scientific community.

A1: We really thank you for reviewing and giving comments for our manuscript. As the reviewer mentioned,  the sample size is the apparent limitation of this study. It might be too small a sample size to generalize our findings to overall mammary tumor as well as human breast cancer. However, we  thought that we might get meaningful results from the biological duplication of the sets of PBMC, CMT and adjacent normal tissues in restricted tumor types and dog breeds. Moreover, we tried to validate our RRBS data using BSC in additional sample sets. We clearly described this limitation and hoped it may help the reviewer and the scientific community have more confidence in our findings. (line 297 at p: 12)

Q2: The method sections of manuscript have not been adequately described. For example in section 4.3, they have performed both DNA and RNA isolation. Why the title is only Genomic DNA Isolation?

A2: M&M section was revised (p: 13-15) and grammar correction was made throughout the revised version of manuscript.

Q3: For the hierarchical clustering and heatmaps (2A, 2B and 4A), what is type of data and method used for clustering is not clear. Again showing only two replicates per group can be perfect compared to showing multiple replicates.

A3: We described it in a figure legend of the revised manuscript. (line 126 at p:4 and line 217 at p:8)

Reviewer 2 Report

Dear Authors

Present manuscript focused a very important topic in veterinary medicine. My concern is related with the poor description of the tumors included in this study, from a clinical and from a histological  point of view (This aspect must be improved). Another concern is related to the fact that authors use only very few number of tumors and all of them from the same histological type. Authors must indicate the limitations of present data taking into account the diversity of other histological tumor types not included in the present manuscript and therefore not analysed.

In table 1 under the title tumor type authors indicate "complex", complex is not a tumor type. and "normal" either a non tumor type. Must correct. 

Some additional references should be consulted and included if authors consider it of relevance.

Semin Oncol. 2017 Aug;44(4):288-300. 

Vet Comp Oncol. 2017 Jun;15(2):655-666.

Vet J. 2015 Aug;205(2):161-74. 

Author Response

Q1: Present manuscript focused a very important topic in veterinary medicine. My concern is related with the poor description of the tumors included in this study, from a clinical and from a histological point of view (This aspect must be improved). Another concern is related to the fact that authors use only very few number of tumors and all of them from the same histological type. Authors must indicate the limitations of present data taking into account the diversity of other histological tumor types not included in the present manuscript and therefore not analysed.

A1: Thank you for your kind comment. We updated our introduction and discussion with clinical information (line 42 at p: 1) and indicated the limitations of this study in the discussion. (line 299 at p:12 and line 349 at p 13)

Q2: In table 1 under the title tumor type authors indicate "complex", complex is not a tumor type. and "normal" either a non tumor type. Must correct.

A2: Tumor type classification in dog was based on classification of WHO mentioned in the revised manuscript.

.

Reviewer 3 Report

The manuscript describes the comparison of DNA methylation between canine mammary tumors and adjacent normal tissue and peripheral blood in dogs with and without mammary tumors. Numbers are minimal making it impossible to assess within-group variation. Thousands of differences are observed including differences linked to genes belonging to relevant functional categories. Overall there is some evidence canine mammary tumors may be a useful model for human breast cancer. 

Major comments: 

I am concerned that in the peripheral blood analysis cell count heterogeneity was not considered and could account for many of the methylation differences (see e.g. PMIDs 22568884 and 24495553). Cell counts based on human references could be used to estimate cell counts in the canine methylation profiles. 

The methods section does not describe specifically what algorithm and parameter settings were used for the DMR analysis.  These should be provided.  The logistic regression model does allow covariates and these could include blood cell count estimates.   

Minor comments: 

“global methylation profiles” 

The term ‘global methylation’ refers to a single estimate of DNA methylation across a genome.  Here however you have measured CpG site-specific methylation so ‘global’ should be replaced with ‘genome-wide’. 

“It is especially known that the complex type of CMT, which includes a myoepithelial cell population and comprises ~20% of total CMT, is a great model for human BC, since myoepithelial cell type BC is 45 rare in humans (~0.1%).” 

Why does such a big difference between human and canine tumors make it a ‘great model for human BC’? 

“Since miRNA has shown several reaction features, it can repress multiple mRNAs.” 

Not clear what ‘shown several reaction features’ means. 

“The number of DMRs found was higher in than PBMC.” 

Higher in what? 

Author Response

The manuscript describes the comparison of DNA methylation between canine mammary tumors and adjacent normal tissue and peripheral blood in dogs with and without mammary tumors. Numbers are minimal making it impossible to assess within-group variation. Thousands of differences are observed including differences linked to genes belonging to relevant functional categories. Overall there is some evidence canine mammary tumors may be a useful model for human breast cancer.

Major comments:

Q1: I am concerned that in the peripheral blood analysis cell count heterogeneity was not considered and could account for many of the methylation differences (see e.g. PMIDs 22568884 and 24495553). Cell counts based on human references could be used to estimate cell counts in the canine methylation profiles.

A1: I appreciated your suggestion. We added supplemental result and discussion regarding the cell population in PBMC (line 308 at p: 12), Supplementary fig. S6.

Q2: The methods section does not describe specifically what algorithm and parameter settings were used for the DMR analysis. These should be provided. The logistic regression model does allow covariates and these could include blood cell count estimates.  

A2: We updated our M&M section (line 407 at p: 14) and made lots of English correction throughout the revised version of manuscript.

Minor comments:

“global methylation profiles”

Q3: The term ‘global methylation’ refers to a single estimate of DNA methylation across a genome.  Here however you have measured CpG site-specific methylation so ‘global’ should be replaced with ‘genome-wide’.

A3: The term ‘global’ was changed to ‘genome-wide.’

Q4: “It is especially known that the complex type of CMT, which includes a myoepithelial cell population and comprises ~20% of total CMT, is a great model for human BC, since myoepithelial cell type BC is 45 rare in humans (~0.1%).” Why does such a big difference between human and canine tumors make it a ‘great model for human BC’?

A4: Myoepithelial cell type breast cancer is very rarely diagnosed in human. It is thus a limitation for further study. Although, there are discrepancies in incidence between human and dog, myoepithelial cell type breast cancer is histologically similar to canine complex adenocarcinoma. (line 50 on p: 2)

Q5: “Since miRNA has shown several reaction features, it can repress multiple mRNAs.”

Not clear what ‘shown several reaction features’ means.

A5: The sentence was corrected (line 75 on p: 2).

Q6: “The number of DMRs found was higher in than PBMC.”

Higher in what?

A6: The sentence was corrected (line 144 on p: 5).

Round 2

Reviewer 2 Report

I have no additional comments to add.